# Untargeted Metabolomics in Forensic Toxicology: A New Approach for the Detection of Fentanyl Intake in Urine Samples

**DOI:** 10.3390/molecules26164990

**Published:** 2021-08-18

**Authors:** Eleonora Amante, Eugenio Alladio, Rebecca Rizzo, Daniele Di Corcia, Pierre Negri, Lia Visintin, Michela Guglielmotto, Elena Tamagno, Marco Vincenti, Alberto Salomone

**Affiliations:** 1Dipartimento di Chimica, Università di Torino, 10125 Torino, Italy; eleonora.amante@unito.it (E.A.); eugenio.alladio@unito.it (E.A.); rebecca.rizzo@unito.it (R.R.); Lia.Visintin@ugent.be (L.V.); alberto.salomone@unito.it (A.S.); 2Centro Regionale Antidoping e di Tossicologia, 10043 Orbassano, Italy; daniele.dicorcia@antidoping.piemonte.it; 3AB SCIEX, Redwood City, CA 01701, USA; pierre.negri@sciex.com; 4Centre of Excellence in Mycotoxicology and Public Health, Faculty of Pharmaceutical Sciences, Ghent University, B-9000 Ghent, Belgium; 5Dipartimento di Neuroscienze Rita Levi Montalcini, Università di Torino, 10126 Torino, Italy; michela.guglielmotto@unito.it (M.G.); elena.tamagno@unito.it (E.T.); 6Neuroscience Institute Cavalieri-Ottolenghi (NICO), 10043 Orbassano, Italy

**Keywords:** metabolomics, forensic toxicology, fentanyl, novel synthetic opioids (NSO), HepG2 cell lines, urine samples, multivariate data analysis, PCA, SIMCA

## Abstract

The misuse of fentanyl, and novel synthetic opioids (NSO) in general, has become a public health emergency, especially in the United States. The detection of NSO is often challenged by the limited diagnostic time frame allowed by urine sampling and the wide range of chemically modified analogues, continuously introduced to the recreational drug market. In this study, an untargeted metabolomics approach was developed to obtain a comprehensive “fingerprint” of any anomalous and specific metabolic pattern potentially related to fentanyl exposure. In recent years, in vitro models of drug metabolism have emerged as important tools to overcome the limited access to positive urine samples and uncertainties related to the substances actually taken, the possible combined drug intake, and the ingested dose. In this study, an in vivo experiment was designed by incubating HepG2 cell lines with either fentanyl or common drugs of abuse, creating a cohort of 96 samples. These samples, together with 81 urine samples including negative controls and positive samples obtained from recent users of either fentanyl or “traditional” drugs, were subjected to untargeted analysis using both UHPLC reverse phase and HILIC chromatography combined with QTOF mass spectrometry. Data independent acquisition was performed by SWATH in order to obtain a comprehensive profile of the urinary metabolome. After extensive processing, the resulting datasets were initially subjected to unsupervised exploration by principal component analysis (PCA), yielding clear separation of the fentanyl positive samples with respect to both controls and samples positive to other drugs. The urine datasets were then systematically investigated by supervised classification models based on soft independent modeling by class analogy (SIMCA) algorithms, with the end goal of identifying fentanyl users. A final single-class SIMCA model based on an RP dataset and five PCs yielded 96% sensitivity and 74% specificity. The distinguishable metabolic patterns produced by fentanyl in comparison to other opioids opens up new perspectives in the interpretation of the biological activity of fentanyl.

## 1. Introduction

In the last few decades, the misuse of opioids in the United States has become a national emergency affecting public health and safety [1]. Although the situation in Europe is not yet alarming, some emerging data is raising concerns. Since 2009, 49 new synthetic opioids (NSO) have been identified, including 34 fentanyl-derivatives. In 2017, approximately 1300 seizures of new opioids were reported to the EU Early Warning System by law enforcement agencies. The majority of these cases (70%) were seizures of fentanyl derivatives, but a number of other types of opioids (such as U-47700 and U-51754) were also reported [2]. Particular concern was raised by the confiscations of carfentanyl, a potent synthetic opioid with an extremely low lethal dose, reported in over 300 seizures in 2017 [2].

To date, fentanyl is one of the most prescribed opioids for the treatment of chronic pain, and several studies revealed that the opioid epidemic originated from inappropriate prescriptions by physicians [3,4,5]. Due to its chemical structure, fentanyl has a highly lipophilic character that allows it to cross the blood–brain barrier more effectively than traditional opioids [6]. Consequently, its potency is estimated to be much higher than morphine [7].

Urine is usually the matrix of choice for drug testing, since it involves non-invasive sampling, well-established procedures, and admissibility. The direct, or targeted, detection of fentanyl in urine is effective as long as the organism does not entirely excrete the specific target substance. However, fentanyl is typically consumed at low doses making the detection of the parent drug and its main metabolite—norfentanyl—challenging. The main limitation for urine testing is its short diagnostic window due to fast metabolism (usually 24–48 h). Thus, urine sampling has to be well-timed, and repeated several times during the observation period to detect the presence of fentanyl and/or its metabolites. Furthermore, the common targeted methods do not reveal any novel/unexpected substance, such as the fentanyl analogues. As a result, targeted methods do not provide a full metabolic profile and therefore require frequent updates to include new targeted substances and consequent re-validation of the analytical methods. To circumvent these limitations, the development of effective approaches for NSO screening may benefit from the recent technological developments of analytical instrumentation and methods [8], including metabolomics. Metabolomics aims to obtain a comprehensive “fingerprint” of the physio-pathological state of an individual in order to discover any anomalous metabolic pattern potentially related to certain diseases or pathological disorders.

Proteomics and metabolomics were introduced in the field of toxicology about 20 years ago, and their application has consistently grown since then [9,10,11]. Currently, few metabolomics studies have been addressed to investigate drug addiction [10,12,13,14], and were mainly focused on traditional drugs of abuse. On the other hand, a few recently published indirect analytical methods based on metabolomics have been developed to detect different drugs [15,16,17]. The findings of these studies demonstrate that the acquisition of a full metabolic profile is often achieved at the expense of the sensitivity toward specific drug metabolites and minor biomarkers. Among the analytical techniques available to investigate the metabolome, liquid chromatography coupled to high resolution mass spectrometry offers several advantages, including high sensitivity, simple sample preparation and broad coverage of small molecules [18,19].

The premise of our study was the hypothesis that the ingestion of fentanyl, even at low doses, may significantly impact the receiving organism, inasmuch as to generate detectable alterations of its metabolomics profile for a longer period than the presence of the drug itself. Secondly, we aimed to investigate whether these hypothetical alterations of the metabolic profile parallel those produced by morphine and other psychotropic drugs or differ from them, under the alternative hypothesis that their biological receptors may vary or not. In vitro models of drug metabolism have emerged as important tools to overcome the limited access to positive urine samples and uncertainties related to the substances taken, the occurrence of poly-consumption, and the discrimination between single-dose and chronic consumption [20]. Experiments performed with cell lines have stable conditions and allow the obtainment of relevant statistical data [21,22].

We initially used an in vitro model system based on the HepG2 hepatocarcinoma cell line to simulate the urinary metabolome of a fentanyl user. HepG2 cell lines allowed us to sequence their analysis at predefined time intervals and under different treatments, and to verify the selectivity of the untargeted approach by testing other psychotropic substances using the same experimental conditions [23]. After the evaluation of in vitro results, urine samples were analyzed to finalize our metabolomics study. Urine was preferred to blood or plasma because of its longer “memory” and wider metabolomics information [24,25]. For the same purpose of maximizing the acquisition of potentially useful input data, sample treatment was minimal (dilute-and-shoot), separation was performed with both reverse phase (RP) and hydrophilic interaction (HILIC) ultra-high performance liquid chromatography (UHPLC), and mass spectrometric analysis was conducted in high resolution with a quadrupole—time of flight mass (QTOF) instrument using the SWATH^TM^ acquisition method [26,27,28].

## 2. Materials and Methods

### 2.1. Chemicals and Reagents

Fentanyl-D5 was produced by Cerilliant (Round Rock, TX, USA); the working solution was prepared at the final concentration of 1.000 ng/mL by dilution with methanol.

Acetonitrile and formic acid were provided by Sigma Aldrich (Milan, Italy). Ultra-pure water was obtained using a Milli-Q^®^ UF-Plus apparatus (Millipore, Bedford, MA, USA).

Fentanyl-D5, norfentanyl-d5, AH-7921-d6, MDMA-d5, cocaine-d3, morphine-d3, nordiazepam-d5, fentanyl, heroin, cocaine, diazepam, ketoprofen, U-47700, and MDMA were obtained from Chiron (Trondheim, Norway, Sigma-Aldrich (Milan, Italy) and the National Institute of Health (ISS). Dulbecco’s modified Eagle’s medium (Sigma-Aldrich, Milan, Italy), glutamine, non-essential amino acids (Sigma-Aldrich, Milan, Italy), bovine fetal serum (FBS) (Sigma-Aldrich, Milan, Italy), antibiotic and antimycotic solution (10,000 units of penicillin, 10 mg of streptomycin and 25 μg amphotericin B per mL) (Sigma-Aldrich, Milan, Italy), β-glucuronidase from Helix Pomatia, HepG2 cell line (American Type Culture Collection, USA), and trypsin-EDTA (Sigma-Aldrich, Milan, Italy) were also obtained.

### 2.2. Samples Collection and Preparation

#### 2.2.1. In Vitro Samples

HepG2, human hepatocarcinoma cells, were maintained in Dulbecco’s modified Eagle’s medium supplemented with 10% fetal-bovine serum, 100 U/mL penicillin, 100 μg/mL streptomycin and 25 μg/mL amphotericin-B.

The number of cells per milliliter was determined using the automated cell counter (TC20, BIO-RAD), and the cells were diluted in 40 mL of growth medium to reach the cell count of 500,000 cells/mL. The cell count was verified again to ensure the final count. 100 μL of this suspension was transferred into each well of 24-well plates, in order to place 50,000 cells per well. After vigorous shaking, the mobility and density of the cells were assessed through observation under the microscope. Finally, the plates were placed in an incubator at 37 °C with 5% CO_2_. After one night, the growth medium was replaced with 250 µL of treatments following the scheme in Figure 1, with each experiment being performed in duplicate (12 treatments × 4 times × 2 repetitions = 96 plates). The treatments were prepared by adding to 10 mL of growth medium in sterile tubes a specific amount of psychoactive substance (i.e., fentanyl, diazepam, U4770, MDMA, heroine, ketoprofen, cocaine) or combination of substances (i.e., fentanyl and heroine) in order to obtain the defined final concentration.

The initial growth medium was replaced with fresh growth medium in the control cells, with no treatment added. The control MeOH treatment consisted in growth medium added with the maximum amount of methanol used to prepare the other treatments. The medium was transferred in an Eppendorf tube at the end of the experimental time points of 1, 3, 24, and 48 h. The two aliquots corresponding to the same sample time in the same plate were combined in the same Eppendorf tube in order to gather approximately 250 μL of each sample for the analysis. All samples were immediately stored at −80 °C and until the time of sample handling and analysis.

The analytical samples were prepared following two different procedures, one comparable to a dilute-and-shoot approach, the second introducing an hydrolysis step with β-glucuronidase from *Helix Pomatia,* developed on the basis of published methods [29,30,31,32,33].

In the first case, the samples were thawed and vortexed for 10 s at low speed. A total of 200 μL of cold acetonitrile was added to 200 μL of sample in order to precipitate the proteins. Then, the sample was vortexed for 10 s and centrifuged for 5 min at 4000 rpm. 350 µL of the supernatant was diluted in 700 µL of the starting mobile phase mixture. An aliquot of 5 μL was directly injected into the UHPLC instrument for both RP and HILIC separations.

In the second procedure, the samples were thawed and vortexed for 10 s at low speed. 10 μL of acetate buffer at pH 5 and 2 μL of β-glucuronidase were added to 100 μL of sample, then the resulting mixture was vortexed for 10 s and placed in the oven at 55 °C for 1 h. In order to stop the enzymatic reaction, 100 μL of cold acetonitrile was added. The sample was then shaken for 10 s and centrifuged for 5 min at 4000 rpm. 175 µL of the supernatant was transferred in a glass tube and the acetonitrile was evaporated under a gentle N_2_ flow at room temperature. The residue was reconstituted with 200 μL of starting mobile phase mixture and an aliquot of 5 μL was directly injected into the UHPLC instrument for both RP and HILIC separations.

Due to the low sample volume available for the analysis, the chance of using a pool of samples as QC was discarded a priori. Since several drugs were tested, a mixture of deuterated drug standards was opted for, consisting of fentanyl-D5, norfentanyl-d5, AH-7921-d6, MDMA-d5, cocaine-d3, morphine-d3, and nordiazepam-d5 at a final concentration of 500 ng/mL, after dilution with the starting mobile phase mixture.

#### 2.2.2. Urine Samples

A total of 81 urine samples were analyzed, including 18 controls and 63 samples that tested positive for at least one psychoactive substance. Among the latter, 18 tested positive to fentanyl and other substances, 6 to fentanyl only, 12 to opioids and 27 to other drugs (including stimulants, cannabinoids, benzodiazepines, etc.). The details are reported in the Appendix A (Appendix A). The samples were made anonymous before performing any interpretation of the measured results. The fentanyl-positive samples considered in this study were obtained from subjects to whom fentanyl was administered in relatively low dosage following severe traumas, often in conjunction with ketamine, morphine and/or benzodiazepines. Another set of subjects were administered only morphine and/or benzodiazepines without fentanyl. Once collected, the urine samples were stored at +4 °C until the routine targeted analyses to confirm the presence of drugs of abuse was completed (typically 1–3 days). The samples were then stored at −20 °C until the untargeted analysis was started. To this end, the samples were thawed out, centrifuged at 13.3× *g* for 5 min, and the supernatant was diluted 1:4 with the starting mobile phase mixture to reach a final volume of 100 μL. An aliquot of 5 μL was directly injected into the UHPLC instrument for both RP and HILIC separations. The quality control (QC) samples were different for RP and HILIC runs. In the former, they consisted of fentanyl-D5 at the final concentration of 500 ng/mL, after dilution with the mobile phase. For the latter, the QC consisted of a pool of the real urine samples.

### 2.3. Preliminary Urine Testing and Confirmation

Urine testing were preliminarily screened via immunoassays and then confirmed using validated UHPLC-MS/MS methods implemented on a SCIEX API 5500 triple quadrupole mass spectrometer operating in the selective reaction monitoring acquisition mode. In particular, the confirmation methods were dedicated to the detection of (i) cocaine, benzoylecgonine (BZ) and cocaethylene, (ii) morphine, codeine, 6-monoacetylmorphine, oxymorphone, oxycodone, buprenorphine, and norbuprenorphine, (iii) 11-nor-delta-9-tetrahydrocannabinol-9-carboxylic acid (THC-COOH), (iv) 7-aminoflunitrazepam, 7-aminonitrazepam, alprazolam, bromazepam, clonazepam, demoxepam, delorazepam, diazepam, flurazepam, hydroxy-flurazepam, flunitrazepam, lorazepam, lormetazepam, mizadolam, nordiazepam, oxazepam, temazepam, and prazepam [34], (v) methadone, EDDP, ketamine, and norketamine, and (vi) synthetic opioids and their metabolites, including fentanyl, norfentanyl, remifentanil, sufentanyl, alfentanyl, tramadol, hydrocodone, carfentanyl, acetilfentanyl, furanilfentanyl, 4-ANNP, and U-47700 [35].

### 2.4. Chromatographic Separation

#### 2.4.1. Reverse Phase (RP) Chromatography

The chromatographic separations were performed using a Shimadzu Nexera 30 UHPLC interfaced to a high-resolution SCIEX X500R QTOF mass spectrometer. A Kinetex C18 column 100 × 2.1 mm, 1.7 μm (Phenomenex) was used for the untargeted separation of the metabolome in the reverse phase mode. The column oven was maintained at 45 °C and the elution solvents were water/formic acid 5 mM (solvent A) and acetonitrile/formic acid 5 mM (solvent B). After an initial isocratic elution at 95% A for 0.5 min, the mobile-phase composition was varied by a linear gradient (A:B, *v*/*v*) from 95:5 to 5:95 in 9.5 min; then the isocratic elution was maintained for 0.5 min. Finally, the initial 95:5 conditions were restored and held for 1.5 min. The flow rate was 0.5 mL/min, and the total run time was 12 min.

#### 2.4.2. Hydrophilic Interaction Chromatography (HILIC)

The HILIC separation was optimized starting from published methods [36,37,38]. For the optimized chromatographic run, a Kinetex HILIC column 100 × 2.1 mm, 1.7 μm (Phenomenex) was used, equipped with a SecurityGuard^TM^ ULTRA (Phenomenex). The column oven was maintained at 45 °C and the elution solvents were ammonium formate 10 mM/formic acid 5 mM (solvent A) and acetonitrile (solvent B). The gradient was optimized as follows: initial isocratic elution at 5% A for 1.5 min, then 25% A was reached in 2.5 min and hold for 0.2 min; 50% A was reached in the following 4 min. Lastly, the initial conditions were restored. The flow rate was 0.5 mL/min and the total run time was 9 min. The order of the samples’ injection was randomized separately for RP and HILIC experiments.

### 2.5. High Resolution Mass Spectrometry Analysis

The high-resolution SCIEX X500R QTOF mass spectrometer (with full width half maximum resolution ≥ 4.2 × 10^4^, measured on the (M+6H)^6+^ charge isotope cluster for bovine insulin at *m*/*z* 296) was operated with an electrospray ion source in positive ion mode with a voltage of 3500 V. The MS/MS data were collected with the SWATH^TM^ acquisition mode [26,39], in which the accumulation windows were optimized using a pool of urine samples. The collision energy (CE) in the collision cell was set at 35 V and the declustering potential (DP) was set at 65 V. The collision energy spread (CES) was also used, to obtain a richer MS/MS spectrum, and set at the value of ±15 V. The intervals of mass scan were 100–1000 *m*/*z* for MS and 50–1000 *m*/*z* for MS/MS, respectively. The MS/MS data relative to real urine samples were not used in the subsequent statistical analysis but were collected for future identification of the relevant biomarkers. The metabolic progression of the psychoactive substances by the HepG2 hepatocarcinoma cells was evaluated using the untargeted MS/MS approach. Simultaneously, the reproducibility of the analyses over the analytical session was checked by evaluating the data collected with the conventional targeted approach. The data were compared with the library HRAM All-in-One v1.1 from LibraryView^TM^ software to detect the deuterated standards contained in the QC, the substances administrated to the HepG2 cell line, and their common metabolites.

The analytical sequence included: injection of the QC to allow the normalization of intra- and inter-batch effects and to verify the retention times reproducibility; injection of the blank solution to clean the column from any potential carry-over effect. Two analytical replicates of each in vitro sample were analyzed, and three technical replicates of each urine sample were injected from the same vial.

### 2.6. Statistical Analysis

The TOF-HRMS raw data files were initially processed using the MarkerView^TM^ software from Sciex. Briefly, background subtraction was performed; then the software identified the significant peaks by selecting only the signals with intensity higher than 100 counts and width approximately equal to 6 s. Subsequently, alignment of the retention times among different chromatographic runs was performed with tolerance of 0.50 on the retention times shift and 5 ppm for the mass. Lastly, a threshold for the number of detected peaks was arbitrarily set to 7000. The resulting peaks tables for the in vitro samples was imported in R Studio, with R version 4.0.2 [40] and SpectrApp [41]. The peak tables obtained from the urine samples was exported in Excel^®^ and then imported in MATLAB—release R2020b and PLS_Toolbox version 8.9.2 [42].

#### 2.6.1. In Vitro Samples

The four datasets collected from the in vitro samples, which were prepared both with and without the hydrolysis step and were analyzed with both RP and HILIC columns, turned out intrinsically homogeneous. For this reason, a simple transformation of the data was performed by dividing the intensity associated with each variable (*m*/*z* values) by the total area of the sample chromatogram; the normalization was performed per row [43]. Several pre-processing methods were tested, namely mean centering, Pareto scaling, and autoscaling, all preceded or not by log-10 transformation [44,45], to obtain the most informative data visualization. Two samples were analyzed for each experimental condition combination (treatment and collecting time), and the results obtained from the resulting two vials were averaged after the normalization.

Principal component analysis (PCA) [46] was applied separately to the RP and HILIC datasets resulting after dilute-and-shoot and hydrolysis preparations to explore the data and to highlight any groups’ inter-correlation of variables and/or features able to differentiate the different treatments. PCA models were calculated using a 10-fold cross-validation approach with venetian blinds sampling design.

#### 2.6.2. Urine Samples

In the present framework, sparse data within the peaks tables were identified (i.e., the peaks detected in less than 15% of the chromatographic runs) and the corresponding peaks were removed from the dataset, decreasing the number of variables from 7000 to 2294 for RP and from 7000 to 3805 for HILIC. Fentanyl, norfentanyl, and all detected direct fentanyl metabolites were discarded from the selected variables.

For RP experiments, the intra-batch and inter-batch normalization of the real samples intensities was accomplished using the area of the fentanyl-D5 peak, detected in all standard runs. For HILIC experiments, the normalization was performed using the chromatographic profiles obtained from the urine pool injection. The areas reported in each variable column were averaged and then used for the normalization. Briefly, the mean of fentanyl-D5 peaks’ area (or pool chromatographic profile) was computed, then the ratio between each fentanyl-D5 peak (or pool chromatographic profile) and its mean value were used to correct the peaks intensity in the corresponding real sample [43].

PCA [46] was subsequently used to verify the reproducibility of the three replicates. In short, the mean-centering pre-processing was independently applied to the areas of the peaks detected in the three technical replicates of each sample. If the three replicates were symmetrically distributed and within (or nearby) the 95% acceptance limits of the model, they were considered as reproducible and the mean of the intensities was computed. This approach offers the advantage of enhancing the exploration of the data structure class by class. The RP and HILIC datasets were then separately explored using PCA models. Several data pre-processing strategies were tested once again.

Lastly, two classification models were built using the soft independent modeling by class analogy (SIMCA) algorithm on RP and HILIC scores, respectively [44,47]. In particular, a one-class SIMCA model was built, using the PCA scores to model the desired group, namely the samples positive to fentanyl. Upon adjustment of the SIMCA parameters, the best performing model was identified as the one providing the lowest root mean square error in cross-validation (RMSECV) [45,48,49]. The acceptability criterion was defined by evaluating the samples positive to fentanyl within the plane consisting of Hotelling’s T^2^ values and Q-residuals. Then, samples from the other classes were projected into the space defined by the SIMCA model [45,50]. The PLS-Toolbox version 8.9.2. [42] and in-home written codes were used to perform these statistical analyses.

## 3. Results and Discussion

### 3.1. Targeted Analysis

The progressive metabolization of the psychoactive substances by the HepG2 cell line was confirmed by observing the targeted MS/MS data obtained using the high-resolution SCIEX X500R QTOF mass spectrometer on the vitro samples. In combination with RP separation, at least one metabolite was detected in each sample for each treatment at every collection time, with the only exception of ketoprofen. On the other hand, metabolites were not identified in several samples when HILIC separation was performed. Controls and quality controls data were also evaluated with the same targeted method. Controls did not show evidence of carry-over or contamination, and quality controls components did not undergo significant changes in measured retention time and exact mass, confirming the reproducibility of chromatographic runs and MS conditions.

From the preliminary targeted confirmation analyses carried out on the urine specimen, four different sample groups could be identified, including: (group 1) negative controls (*n* = 18), (group 2) positive to fentanyl (*n* = 24), (group 3) positive to other opioids (*n* = 12), and (group 4) positive to other drugs (*n* = 27).

Among the 24 urine fentanyl-positive samples, 17 tested positive also to other drugs. In particular, morphine was detected in six samples (25.0%), ketamine and its metabolite norketamine in four samples (16.7%), and benzodiazepines in 10 samples (41.7%). Furthermore, THC–COOH was found in five samples (20.8%) and BZ in only one sample. Details are reported in the Appendix A (Appendix A).

Among the urine samples positive to various opioids, morphine was detected in 8 samples (66.7%), codeine in six samples (50%), buprenorphine and its metabolite norbuprenorphine in three samples (25%). The primary metabolite of heroin, namely 6-MAM, was found in one sample (8.3%) and oxycodone in another one (Appendix A). Among the 27 samples positive to other drugs, 13 were found positive to at least one opioid, too, including morphine, 6-MAM, codeine, and oxycodone (Appendix A).

### 3.2. Examination of Technical Replicates of Urine Samples

The reproducibility of the technical replicates in RP and HILIC analyses was verified by evaluating the PCA scores plots reported in the Appendix A, respectively. Excellent reproducibility was observed for over 90% samples, with the exceptions of sample 3 in RP and sample 6 in HILIC. For reproducible samples, the mean profile was calculated and subsequently interpreted, while outliers 3 and 6 in RP and HILIC respectively were removed and only two datasets were averaged.

For the RP dataset, Pareto scaling prior to PCA model computation turned out the best pre-processing method. For the HILIC dataset, autoscaling proved most appropriate for data visualization. The efficacy of the batch effect normalization was then verified on the mean datasets of final dimensions equal to 81 × 2294 and 79 × 3805 for RP and HILIC, respectively. After identifying each analytical sequence with one color (see Appendix A), it was observed that the samples were randomly distributed along the PCs showing the highest variance, proving that no batch effect was observed in both RP and HILIC datasets.

### 3.3. Data Exploration of Real Samples

#### 3.3.1. In Vitro Samples

Untargeted analysis of the in vitro samples showed no outliers among the duplicate samples and no batch effect. Pareto scaling proved to be the optimal pre-processing strategy. Averaged data from the control samples were subtracted from the other sequences to remove the common contribution of the HepG2 cell line. Four PCA models were built on the samples prepared from dilute-and-shoot or enzymatic hydrolysis procedures, in combination with RP or HILIC separations, respectively. The *m*/*z* signals related to the psychoactive substances used to treat the HepG2 cells were removed from the datasets before building the PCA models. The PCA model calculated on the samples prepared with dilute-and-shoot approach (40 × 295) and RP chromatography provided the most interesting result. The first three PCs explained more than 85% of the variance. In particular, peculiar distributions of the samples are observed by plotting PC2 and PC3 (Figure 2A). The group of fentanyl 50 µM samples and cocaine is isolated from the rest of the samples, while a distinct cluster consists of fentanyl 5 µM, fentanyl 0.5 μM and MDMA, and another cluster is composed by heroin, fentanyl co-administrated with heroin (both 5 µM), and the remaining psychoactive substances (i.e., diazepam, ketoprofen, and U47700). In particular, the PC2 has significant positive loadings for *m*/*z* 353.1141, 166.0849, 355.0609, and 120.0795, mainly describing the cluster consisting of fentanyl 5 µM, fentanyl 0.5 μM and MDMA (Figure 2B). The PC3 component, that principally differentiates the samples related to the 50 μM fentanyl and cocaine samples, provides significant negative loadings values for *m*/*z* 335.1037, 188.0691, 304.1541, 267.1717, 211.1093, 120.0802 and 166.0861. High PC3 loadings values were observed for *m*/*z* 355.0634 and 188.0703.

#### 3.3.2. Urine Samples

After Pareto scaling and control sample subtraction, the variables unequivocally related to the administered psychoactive substances and their direct metabolites were subtracted from the urine samples dataset. Notably, the elimination also included all substances (*m*/*z* values in the data) that turned out significant in the previous PCA models, namely those built from in vitro experiments. The reason for eliminating all in vitro generated metabolites lies in the main objective of the present study which is to investigate the effect of drug intake on the urinary metabolic profile of the recruited subjects, not the metabolism of drug itself. After this data cleaning, four PCA models were built again on the samples prepared with dilute-and-shoot approach or with enzymatic hydrolysis, in combination with RP or HILIC strategies. Actually, the enzymatic hydrolysis proved not to add contribution to the set of significant variables, but rather it enhanced the chemical noise. Overall, the most interesting results were obtained from the dilute-and-shoot approach combined with RP separation. Despite the first three components of the PCA model explain only 52% of the total variance, they are quite informative about the distribution of three groups (i.e., fentanyl, opioids, and other drugs) under exam. In particular, by combining PC2 with PC3, the scores plot reported in Figure 3 is obtained, corresponding to 35% of cumulate variance. PC2 appears to be significantly influenced by the impact that the administration of fentanyl produced on the urinary metabolic profiles under investigation (purple triangles, Figure 3A). Significant loadings values are observed along PC2 and PC3 (Figure 3B), with the remarkable difference that all fentanyl-positive samples are characterized by low PC2 scores, whereas the same samples exhibits PC3 scores widely ranging from highly positive to highly negative. The latter effect can possibly be attributed to the co-administration of other drugs.

Another interesting pattern observed in Figure 3 refers to the potential use of high PC2 scores as a preliminary indicator of traditional opioid/opiates intake. This class includes both traditional opioids (green stars) and “other drugs”, that includes at least one traditional opioid (red circles). The majority of these samples tested positive for morphine and codeine, and few others also tested positive for oxycodone, oxymorphone, methadone, and EDDP. All these samples were mainly characterized by high *m*/*z* 265.1446 and 290.1984 values (Figure 3B).

According to retention times of the significant *m*/*z* loadings, it can be inferred that the poorly separated polar metabolites largely contribute to the PC2 and PC3 significant loadings. In contrast, the non-polar metabolites with elution times higher than 5 min provide virtually no contribution to PC2 and PC3, as their loadings are close to zero.

### 3.4. Construction of Classification Models Based on SIMCA Algorithm for Urine Samples

The unsupervised data exploration accomplished by PCA provided a preliminary statistical tool to discriminate individuals positive to fentanyl from the negative ones, based only on the fraction of their urinary metabolic profile that do not consider the parent drug nor its direct metabolites. This preliminary challenge is accomplished in view of the ultimate goal of detecting fentanyl use/abuse even when both the administered drug and its metabolites are no more detectable in the urine (for example, later than 48 h after intake), as it may occur for potent drugs typically taken at low dosage. To verify this hypothesis further, a supervised approach was tested to build a classification model for prediction. Among the various supervised methods, a single-class modeling SIMCA approach was selected. This approach is particularly well-suited when the acceptability boundaries of a single class (e.g., positive to fentanyl) is set to separate its members from a variety of heterogeneous samples [45,47,50]. Hence, one model for each dataset (RP and HILIC) was built using the samples positive to fentanyl.

For the RP dataset, optimal classification performances were obtained by selecting the 186 variables characterized by a discriminant power higher than 1.5. The lowest RMSECV was obtained from a 4 PCs model, in which only one sample was not recognized as a class member during the cross-validation process (red diamond in Figure 4A). This sample was excluded from the model because it yielded to a T^2^ value higher than the standard acceptance limit, equal to the square root of 2 (Figure 4B). For the remaining 57 samples, relative to no-fentanyl administration, the SIMCA model yielded 42 samples correctly rejected by the class and 15 false positive recognition. The latter included six negative controls, three positives to opioids, and six positives to other drugs, including traditional opioids. It can be concluded that the SIMCA model built on the RP dataset yielded a 96% sensitivity in cross validation and 74% specificity. The prediction results and the residuals plot are reported in Figure 4A,B, where the red dot line represents the acceptance limit of the SIMCA model.

For the HILIC dataset, optimal classification was obtained by setting the discriminant power threshold at 4.5, corresponding to the retention of 390 variables. The lowest RMSECV was obtained by building the SIMCA model with six PCs, resulting in the correct classification in cross-validation of all samples. However, the projection of the 57 test samples from the other three groups resulted in the erroneous inclusion of 14 of them within the class boundaries (Figure 5A,B). Among them, one sample belongs to the class of opioids, seven samples to the class of positive to other drugs (including four positives to at least one traditional opioid), and six samples to the class of negative controls. In this case, the sensitivity and specificity scores in cross-validation were equal to 100% and 76%, respectively. It is noteworthy that most samples positive to opioids other than fentanyl are correctly rejected by both SIMCA models.

It is known that both traditional opioids and NSO act through the activation of the same μ receptors [51,52]. Therefore, their effect on the urinary metabolome could be expected to share several similarities. In contrast, both PCA results and SIMCA classification models of the present study suggest a different and more significant impact of fentanyl on the excreted metabolome. In particular, the two PCA models (typically unsupervised) show very different scores for the two populations (positive to fentanyl and positive to opioids) where the samples positive to opioids are generally grouped with negative controls and samples positive to other drugs—no matter which PC combination is selected, while the samples positive to fentanyl are mostly separated from them. Surprisingly, the number of samples positive to opioids erroneously included in the SIMCA models is even lower than the number of the negative controls accepted within the SIMCA boundaries.

The performance of the two classification models appears quite similar, as they provide comparable percentages of wrong predictions (26% for the RP dataset and 24% for HILIC) and correct identifications (23/24 and 24/24, respectively). However, by matching the SIMCA models obtained from RP and HILIC datasets, it appears that only four samples are misclassified by both models (marked by red arrows in Figure 4A and Figure 5A). This result might suggest that the combination of the two analyses could significantly improve the specificity rate. In theory, a two-steps approach may exploit the HILIC analysis as a surrogate for the confirmation method only on the urinary samples that tested positive by the SIMCA model built on the RP dataset. The false positive rate for the dataset presently used drops from 25% to 7% (4/57 samples). Quite obviously, these are ventured considerations, not supported by sufficient validation. However, they demonstrate that a specific footprint can be detected in the urinary metabolic profile of the subjects to whom fentanyl has been administered, even when all the potential direct drug metabolites (i.e., those identified by in vitro experiments) are excluded from the profile. Further improvements of the sensitivity and specificity scores are expected by refining the variable selection algorithms on significantly larger datasets.

## 4. Study Limitations

The proposed study has two limitations. The first one is related to the lack of a proper external validation of the SIMCA classification models. In fact, the two models have been computed using all the positive samples while no independent validation group was used. This choice was influenced by the limited number of samples, the wide heterogeneity of fentanyl dosage administration (see Appendix A), and the occurrence of combined therapies with other pharmaceuticals. Therefore, it was not possible to exclude some samples from the class modeling process in favor of a test set without losing valuable information. Nevertheless, the unsupervised data exploration on the whole dataset (performed without any variables selection) showed clear separation of the samples positive to fentanyl from the others, opening the way for the application of supervised approaches. In future studies, more samples positive to fentanyl should be independently analyzed and predicted by the two models to exclude the possible occurrence of model overfitting.

The second limitation relates to the lack of urine samples from subjects who consumed fentanyl several days before their collection, when the parent drug and its direct metabolites are no longer detectable in the collected urine sample. Future investigations should verify whether the presented approach is capable of providing a correct classification for these samples. At the proof-of-concept stage, we wanted to explore the potential of an innovative approach, using samples positive to fentanyl, even though at trace level. In fact, 14 samples included in this study had fentanyl urinary concentration levels below 5 ng/mL, and seven of them showed values below 1 ng/mL (the limit of detection of the applied targeted method is equal to 0.2 ng/mL). We believe that the promising results obtained with these low urinary levels from non-chronic users represents a promising projection of the SIMCA models performance on samples with no more residual fentanyl detectable.

## 5. Conclusions

At the present time, forensic toxicology is challenged by the detection of NSO intake which has two main limitations: (i) the potency of these new synthetic drugs is high; hence, the ingested doses are low, making detection of drugs and metabolites challenging in biological fluids, and (ii) the analytical standards are not always or readily available in the laboratory. On the other hand, the high pharmacological effect of NSO is likely to produce a significant metabolic response to their intake. In this study, we aimed to register this urinary metabolic response to fentanyl administration ruling out the potential bias coming from direct fentanyl metabolites, in turn itemized from in vitro HepG2 cell line experiments. The SWATH mass spectra acquisition mode enabled us to detect thousands of compounds excreted, possibly forming the urinary metabolome.

The significant indirect biomarkers revealed by the combination of scores and loadings diagrams obtained from exploratory PCA could not be precisely identified, but they still proved to work as discriminant tools when properly computed together. In the future, the possible identification of these diagnostic indirect biomarkers may possibly lead to the development more sensitive and specific class-targeted methods.

The chance of observing metabolic alterations similar to those produced by fentanyl in subjects exposed to some of its analogues may represent another valuable starting point for future research, since the present untargeted approach is based on the intake effects, making it potentially independent from the taken substance. Anticipated advantages of this strategy include the extension of the detectability window and the possibility of detecting the consumption of substances still unknown or for which the analytical standard is not available. Novel screening methods, that are not directly targeting the chemical structures of the analytes or their metabolites, are critically needed and would help to provide fast response on suspected new drugs consumption.

The observation that the metabolic changes induced by fentanyl ingestion are distinctive with respect to those induced by other opioids opens up new perspectives in the interpretation of the pharmacodynamic profile of fentanyl. The higher potency and increased activity of fentanyl on its receptors represents a possible explanation for the different metabolic cascade.

Several other problems have to be considered when one foresees the routine applicability of the present approach: (i) the metabolomics biomarkers may partly depend on individual features or confounding factors (e.g., genetics, diet, gender), (ii) the ability to readily transfer the method to different laboratories appears challenging, and (iii) the time consuming and expensive procedures involved in metabolomics investigations may reduce their applicability in screening testing. At this stage, the current method needs further studies, improvements, and a comprehensive validation before being adopted and accepted from a legal point of view, with the aim of evaluating whether a perpetrator was or not under the influence of drug. Nevertheless, we envision that further research and technologies development will assist the implementation of untargeted metabolomics in forensic toxicology, overcoming the current multiple testing limitations of the targeted procedures.

## Figures and Tables

**Figure 1 molecules-26-04990-f001:**
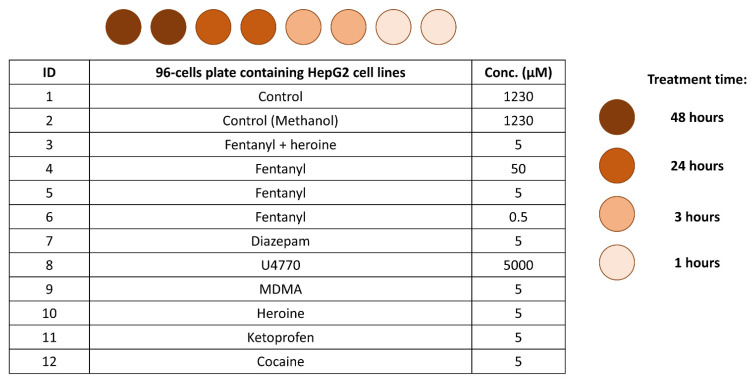
Scheme of the in vitro samples preparation involving 10 psychoactive substance or combination of substances (whose concentration levels are reported in μM), and 4 experimental times for all the psychoactive substances. Samples were prepared in duplicate.

**Figure 2 molecules-26-04990-f002:**
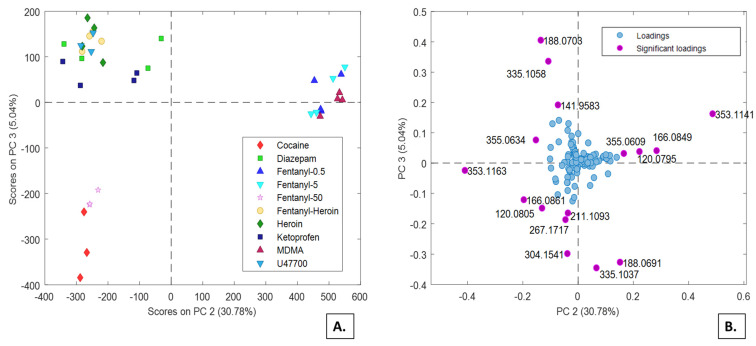
PC2 vs. PC3 scores (**A**) and loadings (**B**) plots from the PCA model calculated on the RP dataset for the HepG2 cell lines treated with several psychoactive substances.

**Figure 3 molecules-26-04990-f003:**
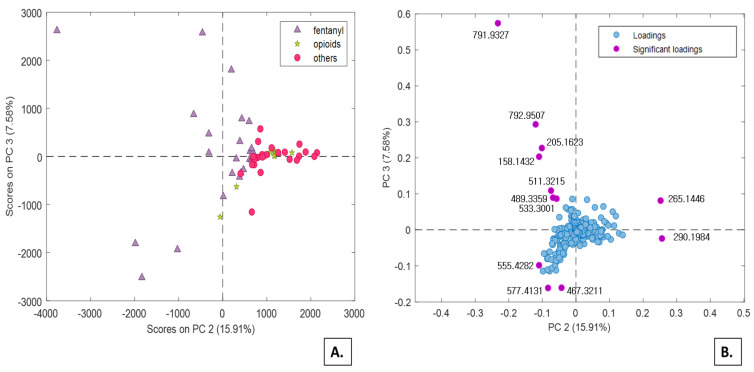
PC2 vs. PC3 scores (**A**) and loadings (**B**) plots from the PCA model calculated on the RP dataset.

**Figure 4 molecules-26-04990-f004:**
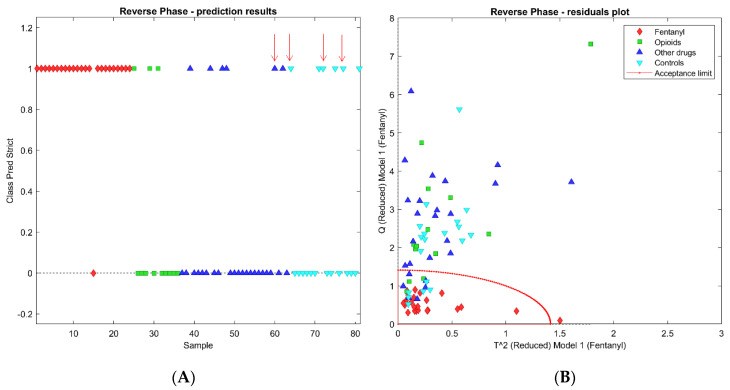
SIMCA models obtained with the RP dataset: (**A**) reports the prediction results and (**B**) the residuals plot. The red arrows indicate the misclassified samples by both the SIMCA models built on RP and HILIC datasets.

**Figure 5 molecules-26-04990-f005:**
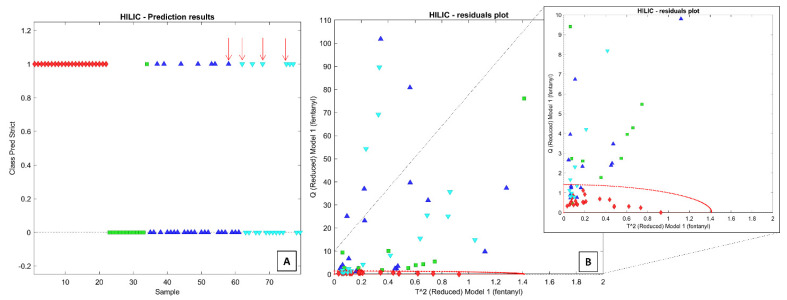
SIMCA models obtained with the HILIC dataset: (**A**) reports the prediction results and (**B**) the residuals plot. The red arrows indicate the misclassified samples by both the SIMCA models built on RP and HILIC datasets.

## Data Availability

Data will be made available upon request to the authors.

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
