# Peer review of "Untargeted Metabolomics in Forensic Toxicology: A New Approach for the Detection of Fentanyl Intake in Urine Samples"

_molecules, 2021, doi:10.3390/molecules26164990_

Round 1

Reviewer 1 Report

This manuscript by Salomone and coworkers describes their efforts in developing a new approach for the detection for the fentanyl intake in urine using both UHPLC reverse phase and HILIC chromatography combined with QTOF mass spectrometry. Although there are some limitations as also pointed out in the manuscript, I believe the approach presented in this manuscript would be useful towards the detection of fentanyl and NSO. I recommend publishing after some minor revisions:

  1. Page 4, line 137, Please double check this sentence. I am not sure what this means “Error! Reference source not found.”
  2. Figure 4A, what are these four 4-arrows for?
  3. Page 15, line 494-495, “namely n. 61, 65, 73, and 78 (marked by red arrows in 494 Figures 4A and 5A). “ Change “n.” to “no.” Are these the correct sample numbers? In the Figure 4A and 5A, it looks like the “blue diamond” that the first red arrow pointing to are smaller than sample no. 60. These need to be double checked.

Author Response

Reviewer #1

This manuscript by Salomone and coworkers describes their efforts in developing a new approach for the detection for the fentanyl intake in urine using both UHPLC reverse phase and HILIC chromatography combined with QTOF mass spectrometry. Although there are some limitations as also pointed out in the manuscript, I believe the approach presented in this manuscript would be useful towards the detection of fentanyl and NSO. I recommend publishing after some minor revisions:

  1. Page 4, line 137, Please double check this sentence. I am not sure what this means “Error! Reference source not found.”

The typo “Error! Reference source not found.” has been corrected with the following sentence After one night, the growth medium was replaced with 250 µL of treatments following the scheme in Figure 1, with each experiment being performed in duplicate (12 treatments × 4 times × 2 repetitions = 96 plates).”

  1. Figure 4A, what are these four 4-arrows for?

The four arrows in Figures 4A and 5A indicate the common misclassified samples by both the SIMCA models built on the RP and HILIC datasets. This indication is remarked in the text (lines 500-502), as follows: However, by matching the SIMCA models obtained from RP and HILIC datasets, it appears that only four samples are misclassified by both models (marked by red arrows in Figures 4A and 5A).”

However, as suggested by the reviewer, a brief description of the red arrows has been added to the figure captions of Figures 4 and 5: “The red arrows indicate the misclassified samples by both the SIMCA models built on RP and HILIC datasets.”

  1. Page 15, line 494-495, “namely n. 61, 65, 73, and 78 (marked by red arrows in 494 Figures 4A and 5A). “ Change “n.” to “no.” Are these the correct sample numbers? In the Figure 4A and 5A, it looks like the “blue diamond” that the first red arrow pointing to are smaller than sample no. 60. These need to be double checked.

The samples were correctly indicated in the text. However, as suggested by the reviewer, the number of the samples have been removed from the text in order to avoid the readers any misunderstanding; in fact, their notation did not provide any further information to the readers.

Reviewer 2 Report

The “indirect method” of detection of fentanyl intake described by Amante et al. has a number of limitations (drawbacks). However, most of them have been clearly described in the manuscript, which is greatly appreciated, therefore, I suggest to accept the manuscript after a minor revision.

The question is if such a method can be acceptable from the legal point of view (e.g. to prove if the perpetrator was or not under the influence of drug), a brief discussion should be added.

If I understood it correctly, the “metabolomic fingerprints” were obtained in the positive ion mode. The question is, if in the negative ion mode the metabolomic image would be less complicated, thus the methods would be relatively easier. The brief justification of the choice of the positive ion mode should be added.

What about reference 25 (line 647)? Do the references 36 and 37 really concern the hydrophilic interaction chromatography (line 225)? Do the reference 21 and 38 can be replaced by the more appropriate ones? I strongly suggest the authors carefully check if the reference cited are appropriate.

In the title - “….the detection of fentanyl intake in urine” is it correctly expressed?

Line 137 - “Error! Reference source not found.” (?)

Figure 1 – it is not clear which ID corresponds to the which treatment time.

Line 379-382 - the sentence is not clear. What the authors mean by the “cluster”?

Line 373-375 –  the sentence is awkward, should be rewritten (formally upon ESI condition we deal with protonated molecules not with molecular ions).

The manuscript should be rather submitted to the Analytical Chemistry section, not to the Medicinal Chemistry section as it was.

Author Response

Reviewer #2

The “indirect method” of detection of fentanyl intake described by Amante et al. has a number of limitations (drawbacks). However, most of them have been clearly described in the manuscript, which is greatly appreciated, therefore, I suggest to accept the manuscript after a minor revision.

The question is if such a method can be acceptable from the legal point of view (e.g. to prove if the perpetrator was or not under the influence of drug), a brief discussion should be added

As it was reported in the conclusions of the original manuscript, we totally agree with the reviewer: “Several other problems have to be considered when one foresees the routine applicability of the present approach: i) the metabolomics biomarkers may partly depend on individual features or confounding factors (e.g., genetics, diet, gender), ii) the ability to readily transfer the method to different laboratories appears challenging, and iii) the time consuming and expensive procedures involved in metabolomics investigations may reduce their applicability in screening testing. Nevertheless, we envision that further research and technologies development will assist the implementation of untargeted metabolomics in forensic toxicology, overcoming the current multiple testing limitations of the targeted procedures.”.

Moreover, to fulfil the correct indications of the reviewer, the following sentence has been added: “At this stage, the current method needs further studies, improvements, and a comprehensive validation before being adopted and accepted from a legal point of view, with the aim of evaluating whether a perpetrator was or not under the influence of drug.”

If I understood it correctly, the “metabolomic fingerprints” were obtained in the positive ion mode. The question is, if in the negative ion mode the metabolomic image would be less complicated, thus the methods would be relatively easier. The brief justification of the choice of the positive ion mode should be added.

The decision of evaluating the metabolomic fingerprint by using a positive ion mode was made according to literature and preliminary studies performed in our laboratory. For these reasons, we believe no further details should be added in the text since a comprehensive discussion of the obtained results is reported in the text.

What about reference 25 (line 647)? Do the references 36 and 37 really concern the hydrophilic interaction chromatography (line 225)? Do the reference 21 and 38 can be replaced by the more appropriate ones? I strongly suggest the authors carefully check if the reference cited are appropriate.

As correctly suggested by the reviewer, the references were carefully checked and updated.

In the title - “….the detection of fentanyl intake in urine” is it correctly expressed?

The title was modified to fulfil the suggestion of the reviewer and avoid any misunderstanding, as follows: “Untargeted metabolomics in forensic toxicology: a new ap-proach for the detection of fentanyl intake in urine samples”

Line 137 - “Error! Reference source not found.” (?)

The typo “Error! Reference source not found.” has been corrected with the following sentence After one night, the growth medium was replaced with 250 µL of treatments following the scheme in Figure 1, with each experiment being performed in duplicate (12 treatments × 4 times × 2 repetitions = 96 plates).”

Figure 1 – it is not clear which ID corresponds to the which treatment time.

To ease the interpretation of Figure 1, the figure caption has been modified, as follows: “Figure 1. Scheme of the in vitro samples preparation involving 10 psychoactive substance or combination of substances (whose concentration levels are reported in μM), and 4 experimental times for all the psychoactive substances. Samples were prepared in duplicate.”

Line 379-382 - the sentence is not clear. What the authors mean by the “cluster”?

The term “cluster” indicates a group of samples that are arranged close to each other in the hyperspace calculated by the multivariate models. Since this term is largely used, we believe no further explanations should be added in the text

Line 373-375 –  the sentence is awkward, should be rewritten (formally upon ESI condition we deal with protonated molecules not with molecular ions).

As correctly suggested by the reviewer, the sentence has been modified as follows: “The m/z signals related to the psychoactive substances used to treat the HepG2 cells were removed from the datasets before building the PCA models.”

The manuscript should be rather submitted to the Analytical Chemistry section, not to the Medicinal Chemistry section as it was.

We let the Editor decide about this suggestion.